# Vaginal Lactobacilli and Vaginal Dysbiosis-Associated Bacteria Differently Affect Cervical Epithelial and Immune Homeostasis and Anti-Viral Defenses

**DOI:** 10.3390/ijms22126487

**Published:** 2021-06-17

**Authors:** Sabrina Nicolò, Michele Tanturli, Giorgio Mattiuz, Alberto Antonelli, Ilaria Baccani, Chiara Bonaiuto, Simone Baldi, Giulia Nannini, Marta Menicatti, Gianluca Bartolucci, Gian Maria Rossolini, Amedeo Amedei, Maria Gabriella Torcia

**Affiliations:** 1Department of Experimental and Clinical Medicine, University of Firenze, 50134 Florence, Italy; sabrina.nicolo@unifi.it (S.N.); michele.tanturli@unifi.it (M.T.); giorgio.mattiuz@unifi.it (G.M.); alberto.antonelli@unifi.it (A.A.); ilaria.baccani@unifi.it (I.B.); chiara.bonaiuto@libero.it (C.B.); simone.baldi@unifi.it (S.B.); giulia.nannini@unifi.it (G.N.); gianmaria.rossolini@unifi.it (G.M.R.); 2Clinical Microbiology and Virology Unit, Careggi University Hospital, 50139 Florence, Italy; 3Department of Neurosciences, Psychology, Drug Research and Child Health, Section of Pharmaceutical and Nutraceutical Sciences, University of Firenze, 50134 Firenze, Italy; marta.menicatti@gmail.com (M.M.); gianluca.bartolucci@unifi.it (G.B.)

**Keywords:** vaginal microbiota, *Lactobacillus*, *Gardnerella vaginalis*, IFN-γ, IL-17

## Abstract

Persistent infection with High Risk-Human Papilloma Viruses (HR-HPVs) is a primary cause of cervical cancer worldwide. Vaginal-dysbiosis-associated bacteria were correlated with the persistence of HR-HPVs infection and with increased cancer risk. We obtained strains of the most represented bacterial species in vaginal microbiota and evaluated their effects on the survival of cervical epithelial cells and immune homeostasis. The contribution of each species to supporting the antiviral response was also studied. Epithelial cell viability was affected by culture supernatants of most vaginal-dysbiosis bacteria, whereas *Lactobacillus gasseri* or *Lactobacillus jensenii* resulted in the best stimulus to induce interferon-γ (IFN-γ) production by human mononuclear cells from peripheral blood (PBMCs). Although vaginal-dysbiosis-associated bacteria induced the IFN-γ production, they were also optimal stimuli to interleukin-17 (IL-17) production. A positive correlation between IL-17 and IFN-γ secretion was observed in cultures of PBMCs with all vaginal-dysbiosis-associated bacteria suggesting that the adaptive immune response induced by these strains is not dominated by T_H_1 differentiation with reduced availability of IFN-γ, cytokine most effective in supporting virus clearance. Based on these results, we suggest that a vaginal microbiota dominated by lactobacilli, especially by *L. gasseri* or *L. jensenii*, may be able to assist immune cells with clearing HPV infection, bypasses the viral escape and restores immune homeostasis.

## 1. Introduction

Microorganisms of the “cervico-vaginal microbiota” establish mutual relationships with the host and strongly contribute to defending the mucosal barrier against the invasion of sexually transmitted pathogens, including the Human Papilloma Virus (HPV). *Lactobacillus* species, evolutionally selected in the vaginal environment, are mainly represented by *Lactobacillus crispatus*, *Lactobacillus gasseri*, *Lactobacillus iners* and *Lactobacillus jensenii* and contribute to the host’s defenses against invading pathogens by lowering the pH (through the production of lactic acid) and secreting huge amounts of antimicrobial peptides (AMPs) [1]. *Lactobacillus* species dominate the vaginal microbiota in a large majority of women and define specific vaginal microbial communities [2,3]. In detail, *L. crispatus* is representative of community state type (CST)-I, *L. gasseri* of CST-II, *L. iners* of CST-III and *L. jensenii* of CST-V, respectively [2,3,4]. Loss of the *Lactobacillus* dominance and the colonization by anaerobic and aerobic species define the CST-IV vaginal microbiota. Strains of the genera *Gardnerella*, *Atopobium*, *Prevotella*, *Megasphaera*, *Mobiluncus*, *Streptococcus* and *Ureaplasma* are highly represented in CST-IV, with *Gardnerella vaginalis* usually representing the dominant species. This bacterial community produces lower amounts of AMPs and lactic acid and represents a condition of vaginal dysbiosis highly associated with bacterial vaginosis (BV), the most common bacterial infection of the lower female genital tract [5,6,7].

Vaginal-dysbiosis-associated bacteria often produce mucin-degrading enzymes [3,8] and induce a pro-inflammatory response [8] with impairment of mucosal barrier that facilitates invasion by sexually transmitted pathogens, including high-risk (HR) Human Papilloma Virus (HPV) [3,4]. Persistent infection with HR-HPVs occurs in 10% of infected women and is a primary cause of cervical cancer worldwide [9].

The antiviral-specific immune response is crucial to the eradication of HPV infection and requires the cooperation of CD4^+^ T-helper (T_H_) and cytotoxic CD8^+^ T cells (CTLs) [10]. In fact, the high levels of IFN-γ secreted by T_H_1 cells potentiate the cytotoxic activity of CTLs that specifically recognize and kill cells expressing viral antigens linked to MHC I molecules [11]. Clinical studies confirmed the strong association between the T_H_1 pattern and the clearance of HR-HPV infection [12]. In contrast, IL-17 has been shown to suppress the effectors of the immune response in HPV-associated diseases [13], and the role of T_H_17 and IL-17 seems more crucial in immune enhancement and disease progression but not in the eradication of HR-HPV infection [14].

HR-HPVs evolved different mechanisms of escape from host adaptive response, secreting lower amounts of proteins and manipulating the antigen processing machinery [15]. Despite this, infection clearance is not a rare event and is often associated with the composition of vaginal microbiota [3,16].

Vaginal microbiota composition is essential in preserving the integrity of the cervical epithelium and the functions of the cervical barrier against the invasion of sexually transmitted pathogens [3]. The production of lactate by *Lactobacillus* species maintains the vaginal pH at 3.5–4.5, preventing the overgrowth of opportunistic pathogens and maintaining low concentrations of Short Chain Fatty Acids (SCFAs). Differently than the gastrointestinal tract, an increase in vaginal SCFAs and a concomitant decrease in lactate is always a marker of dysbiosis, a condition that allows pathogens to propagate ascending intrauterine infection with adverse reproductive outcomes, including preterm birth [17]. Due to the high concentration of lactate, the concentration of SCFAs is usually low in *Lactobacillus*-dominated microbiota. An increase in vaginal SCFAs is always a marker of dysbiosis, a condition that allows infections by pathogenic microorganisms [1].

Vaginal bacteria also regulate the functions of antigen-presenting cells (APCs) and the activation of memory T_H_1, T_H_17 and regulatory T (T_reg_) lymphocytes in the submucosal compartment [18]. Activation of these cells may result in high concentrations of IFN-γ, IL-17 and IL-10 that can support or repress the host’s specific response against invading pathogens, including HR-HPV [12,13]. Vaginal-dysbiosis-associated bacteria were repeatedly linked with persistent HR-HPV infection and with cervical cancer [19,20]. However, the contribution of each species in epithelial cell damage and the antiviral response has not been fully defined.

Here, we selected strains of bacterial species representative of each vaginal CST and evaluated their effects on the production of inflammatory cytokines by HPV-transformed cervical epithelial cells and peripheral blood mononuclear cells (PBMCs) from healthy donors.

## 2. Results

### 2.1. Bacterial Strains and Analysis of Short Chain Fatty Acids (SCFAs) Production

Strains of *L. crispatus*, *L. gasseri*, *L. iners* and *L. jensenii* were used as representatives of CST-I, II, III and V, respectively. *G. vaginalis*, *Atopobium vaginae*, *Megasphaera micronuciformis* and *Prevotella bivia* were used as representatives of CST-IV. Table 1 shows the related features of the used strains.

SCFAs have been reported to affect epithelial cell metabolism and innate inflammatory response [20,21]. Acetate, propionate, butyrate and isovalerate are usually found in the vaginal environment [22]. As the first step of our study, we assessed the production of SCFAs in culture supernatants of all bacterial strains by gas-chromatography coupled with a mass spectrometry (GC-MS) system.

Acetic acid turned out to be the dominant SCFA produced by all vaginal bacteria. Figure 1A shows that all vaginal lactobacilli produced a lower acetic acid amount than *G. vaginalis* that usually represents the dominant species of vaginal dysbiosis.

On the other hand, vaginal lactobacilli produced butyric and valeric acid amounts higher than *G. vaginalis* strains. Figure 1B shows that the metabolic profile of *A. vaginae* and *P. bivia* is largely superimposable to that of *G. vaginalis*. In contrast, *M. micronuciformis* produced low amounts of either acetic and butyric acid, and its metabolic profile was intermediate between lactobacilli and *G. vaginalis* strains.

### 2.2. Effects of Supernatants and Bacterial Lysates on Viability of Cervical Epithelial Cell Lines

The cervical epithelial cell lines SiHa and CaSki were cultured in the presence or absence of different amounts of bacterial culture supernatants or bacterial lysates for 24 h. Intracellular ATP was measured as an index of metabolically active cells.

Figure 2A shows that supernatants from all lactobacilli cultures induced a significant increase in ATP production by SiHa cells. In contrast, supernatants from most vaginal dysbiosis bacteria affected cell viability in a dose-dependent manner. The greatest decrease in cell viability was observed in cultures with supernatants from *A. vaginae* culture. Lysates from vaginal dysbiosis bacteria and from *L. iners* induced a decrease in the viability of SiHa cells (20–50%).

The effects of bacterial supernatants on CaSki cell viability were comparable to those induced in SiHa cells (data not shown). Bacterial lysates from *G. vaginalis*, *P. bivia* and from *L. jensenii* significantly affected CaSki cell viability (Appendix A).

The whole data suggest that most vaginal lactobacilli best support the viability and metabolic activity of cervical epithelial cells. In contrast, vaginal dysbiosis bacteria produce factors that affect the viability of epithelial cells and potentially compromise the cervical barrier.

### 2.3. Cytokine’s Production by CaSki and SiHa Cells Cultured with Vaginal Bacteria

SiHa and CaSki cells were cultured with heat-inactivated bacteria (50 MOI/cell) from each species representative of the vaginal microbiota, and the concentration of TNF-α, IL-1β and IL-8 was assessed after 5 days of stimulation. The data obtained have shown that no significant increase in these cytokines compared to unstimulated control was evident in bacterial stimulated cultures (Appendix A). We noted, however, that, despite the huge amounts of IL-8 that were spontaneously produced by CaSki cells, vaginal dysbiosis bacteria slightly increase the production of this cytokine, while lactobacilli always led to a decrease (12–30%) (Appendix A).

### 2.4. Cytokine’s Production by PBMCs Stimulated with Dominant Species of Vaginal Microbiota

As T_H_1 differentiation and IFN-γ production are essential to eradicate viral infections [12], we assessed whether vaginal *Lactobacillus* species induce the production of IFN-γ differently than vaginal dysbiosis bacteria. We cultured PBMCs from healthy donors (*n* = 9) with heat-inactivated bacteria (50 MOI/cell) from all strains reported in Table 1, and cytokine concentration was assessed in supernatants after 5 days of culture. In addition to IFN-γ, the production of IL-4, IL-17, IL-10, IL-6, IFN-γ and IP-10 was analyzed. Figure 3 shows that almost all species of vaginal microbiota were able to induce the production of a significant amount of IFN-γ, IL-17, IL-6 and IL-10 compared to unstimulated controls. In contrast, they did not induce significant production of IFN-α, IP-10 and IL-4 (Appendix A).

Among lactobacilli, *L. gasseri* was the species that induced the highest amounts of IFN-γ and the lowest amounts of IL-17; *L. iners,* in contrast, was an optimal stimulus for either IFN-γ and IL-17 production.

*G. vaginalis* strains induced more IL-17 production compared to lactobacilli, and significant differences emerged, especially from the amounts induced by *G. vaginalis* strains and those induced by *L. gasseri*.

The anti-inflammatory cytokine IL-10 was significantly induced by all bacterial strains compared to unstimulated control, and in particular, *Lactobacillus* species induced amounts of IL-10 highest than *G. vaginalis* or *P. bivia*. However, statistical analysis did not reveal significant differences among the bacterial-stimulated cultures.

To further confirm the differences in cytokine’s production between lactobacilli and vaginal dysbiosis bacteria, we obtained pools of bacterial lysates from (1) *Lactobacillus* species (*L. crispatus*, *L. jensenii*, *L. gasseri* and *L. iners*); (2) *G. vaginalis* strains (315-A, 49145, 14019 and 14018); (3) other vaginal dysbiosis bacteria (*A. vaginae*, *P. bivia* (DNF-188, DNF-650) and *M. micronuciformis*) and used them to stimulate PBMCs as reported above. Culture supernatants were collected after 5 days of stimulation, and the production of IFN-γ, IL-17 and IL-10 was assessed.

Figure 4 shows that IFN-γ production induced by vaginal lactobacilli was significantly higher than those induced by *G. vaginalis*. In contrast, *G. vaginalis* promoted higher production of IL-17 and lower amounts of IL-10 compared to lactobacilli as well as to other species of vaginal dysbiosis.

Finally, Spearman’s correlation analysis among the cytokine levels in cultures with vaginal dysbiosis bacteria revealed a significant correlation between the production of IFN-γ and that of IL-17 (Figure 5 and Appendix A). Notably, these data suggest that vaginal dysbiosis bacteria may compromise the anti-viral T_H_1 response by increasing the differentiation of an increased number of T_H_17 cells and IL-17 concentration in the vaginal environment.

## 3. Discussion

Epithelial and immune homeostasis in the vaginal microenvironment is crucial for host defenses against sexually transmitted pathogens [21,22,23,24,25]. Microbial and the host metabolites in the host microenvironment may affect the course of sexually transmitted infections [26].

We observed that products from most vaginal-dysbiosis-bacteria affected the cell viability of cervical epithelial cells in a dose-dependent manner while products from all lactobacilli (either secreted or not secreted) are able to maintain or increase cell viability. A single exception was observed with lysates from *L. iners* that induced a decrease in the viability of SiHa cells. Although we are aware that in vitro culture of cervico-vaginal epithelial cells is not representative of the complexity of the vaginal microenvironment, our data support the hypothesis that vaginal dysbiosis bacteria and, to a lesser extent, *L. iners* compromise the cervical epithelial barrier. Similar data were reported by Anton L. et al. [27], Randis T. et al. [28] and Lopez-Moreno et al. [29].

Following bacterial stimulation, the cervical epithelial cells produce IL-8, a potent chemoattractant and activator of polymorphonuclear leukocytes [30,31]. Huge amounts of this chemokine are spontaneously produced by HPV-transformed cervical epithelial cell lines [32]. We found that *L. crispatus* and *L. gasseri* were able to negatively interfere with the molecular pathways leading to the high constitutive production of IL-8. In contrast, the *G. vaginalis* increased chemokine production.

The high production of lactate and low production of SCFAs by lactobacilli could be responsible for the modulation of pro-inflammatory properties of cervicovaginal epithelial cells [17].

In addition, the results from the SCFAs profile, performed in supernatants from bacterial cultures, show that *Lactobacillus* species produced a much lower amount of the pro-inflammatory acetic acid compared to all vaginal-dysbiosis-associated bacteria. In contrast, and according to other reports, lactobacilli produced higher amounts of butyric and valeric acid, metabolites with anti-inflammatory activity [33].

A metabolomic analysis performed on vaginal samples from HPV+ and HPV- women ascertained that the metabolome of vaginal-dysbiosis-bacteria clustered differently from *Lactobacillus*-dominated microbiota [26].

To summarize, our data are in accordance with previous reports showing that most of the vaginal dysbiosis bacteria affect the viability and the inflammatory properties of cervical epithelial cells [34,35] and may therefore contribute to increasing the risk of sexually transmitted viral infections, including HR-HPV infection.

HR-HPVs are spontaneously cleared by the immune response in most infections, but viral persistence occurs in 10% of infected women and may induce carcinogenesis [9,36]. Impairment of the vaginal epithelial barrier, chronic inflammation, alterations of the metabolic signaling and of the immune response are all involved in carcinogenesis [37].

The clearance of HPV infections is associated with an optimal level of IFN-γ produced by the T_H_ cells [38]. The importance of immune homeostasis is highlighted by immunosuppressed women who display the increase in the incidence and persistence of HR-HPVs infections [39]. Moreover, prolonged inflammatory response and high secretion of exosomes in the vaginal environment may promote the progress of intraepithelial lesions [37].

The vaginal microbiota compositions affect the rate of infection as well as its outcome, and the CST-IV microbiota profile, dominated by common vaginal dysbiosis bacteria, has emerged as a risk factor for persistent HPV infection [3,16]. However, women who recovered from HPV infection show a significant reduction of CST-IV and an increase in CST-I compared to the time of recruitment [40]. On the other hand, *L. gasseri* (CST-II) has been associated with the complete clearance of the virus [41]. Although the number of enrolled patients was rather limited in both reports, these data suggest that vaginal *Lactobacillus* species do not have comparable protective effects against HR-HPV infections.

In addition, it is not known whether and how vaginal bacteria affect the adaptive immune response to HR-HPV infection and so the viral clearance.

In fact, in literature, most of the studies are focused on the pro-inflammatory or immunomodulating activity induced by vaginal bacteria interacting with cervical epithelial cells or with cells of the innate immunity [34,35]. These important studies have established a significant correlation between vaginal dysbiosis bacteria (CST-IV) and the progression of HPV infections to preneoplastic (CIN1/2) or neoplastic (CIN 3) stages [19,33].

The connection between the persistence of infection and the neoplastic progression was represented by the inflammatory potential of vaginal bacteria [42,43]. Bacterial species of genera *Ureaplasma*, *Atopobium*, *Prevotella*, *Gardnerella*, *Sneathia*, and especially, *Fusobacteria* have been associated with an increased oncogenic risk [41,44]. Among lactobacilli, *L. iners* can promote the progression of infection [45,46].

A more in-depth investigation of the relationship between vaginal bacteria, immune response and persistent infection and, in detail, how some vaginal bacteria can affect the adaptive immune response was performed by van Teijlingen et al. [47]. The authors studied the effect of two species often reported in vaginal dysbiosis CST-IV, *Megasphaera eldsenii* and *Prevotella timonensis,* on the activation of dendritic cells (DCs) and compared their effects with those induced by *L. crispatus*. They found that *P. timonensis* induces a strong T_H_1 response while *L. crispatus* and *M. elsdenii* did not affect T_H_ polarization. A different study reports that *L. crispatus* confers an anti-inflammatory phenotype to DCs through up-regulation of anti-inflammatory/regulatory IL-10 cytokine production and induction of T_regs_ at optimal dosage [48]. Finally, Mitra et al. reported that the depletion of *Lactobacillus* species and the presence of anaerobic taxa of genus *Gardnerella*, *Megasphaera* and *Prevotella* are associated with persistence and slower regression of CIN2 lesions [49].

Previous studies observed that the production of IFN-γ following HPV 16 peptide stimulation is higher in recovered or HPV-negative women compared to those identified in cervical intraepithelial lesions (CIN) [50]. Ondondo and co-workers recently reported that men with HPV clearance had significantly higher IFN-γ levels than those with persistent HPV infection [51]. These data underline the relevance of T_H_1 cell-mediated cytokine response in HPV clearance, but they do not define the role of each bacterial species in supporting the antiviral response.

Our data show that all vaginal bacteria induce the production of IFN-γ, with *L. gasseri* being the best inducer of the cytokine. The differences among *Lactobacillus* and vaginal dysbiosis species mostly reside in the bacteria’s ability to stimulate T_H_17 differentiation and the production of IL-17 at the same time. In this scenario, *G. vaginalis* strains induce the production of greater amounts of IL-17 compared to lactobacilli suggesting that the adaptive immune response induced by these bacteria is not dominated by T_H_1 differentiation and that the combined effects of lower IFN- γ availability and higher IL-17 concentration does not appropriately support the specific antiviral response. In support of this concept, a significant correlation between the production of IFN-γ and IL-17 emerged for all CST-IV species but not for lactobacilli. This suggests that T_H_17 differentiation is induced as well as that of T_H_1 cells and potentially compromises the antiviral response, which does not benefit from T_H_17 effectors [12,13,14,52]. Moreover, IL-17 may be crucial in immune enhancement and disease progression.

In accordance with our results, Gosmann et al. observed an increase in the numbers of activated mucosal T_H_ cells in the concentrations of IL-17 and IL-17-inducing cytokines (IL-23 and IL-1β) in the cervicovaginal lavage obtained from women with CST-IV dominated microbiota [13].

## 4. Materials and Methods

### 4.1. Bacterial Strains

A collection of 12 bacterial reference strains were included in the study, and related features are reported in Table 1. *L. crispatus* (JV-V01), *L. gasseri* (SV-16A), *L. iners* (UPII-60-B) and *L. jensenii* (JV-V16) were used as representative of CST-I, II, III and V, respectively. *G. vaginalis* was selected as representative of CST-IV and four strains isolated respectively from healthy women (315-A) or from women with bacterial vaginosis with Nugent Score 5 (49145/JCP-7276), 8 (14019/JCP-7659), 10 (14018/JCP-7275) were selected. *A. vaginae* (DSM-15829), *M. micronuciformis* (DNF00954) and *P. bivia* (DNF 00188 and DNF-00650) were also used as representative of CST-IV.

### 4.2. Bacterial Cultures

Anaerobic bacteria were grown in Tryptic Soy Agar (TSA), composed by Tryptic Soy Broth (Oxoid, Basingstoke, UK) and 15mg/L of Bacto-Agar (Sigma Aldrich, St. Louis, MO, USA), with the addition of 5% Horse lysed whole-blood (Oxoid, Basingstoke, UK). The plates were incubated at 37 °C for 72 h in anaerobic conditions inside a jar (AnaeroGen™, Thermo Fisher Scientific, Waltham, MA, USA) to create ideal growth conditions (CO_2_: 9–13.0%).

Bacterial strains were also grown in liquid cultures using TSB with 5% horse lysed blood (Oxoid, Basingstoke, UK). The turbidity of the culture broth of each individual strain was measured by using the DensiCHECK densitometer after centrifuging 1 mL of culture at 4000× *g* for 5 min and resuspending the pellet in 1 mL of physiological solution.

Bacterial concentration was calculated according to the following formula:[*bacteria*l] = [*McFarland* ∗ 1.5/0.5] ∗ 10^8^;

Optical Density assessment (OD) was performed using DensiCHECK™ densitometer (bioMérieux, Marcy l’Étoile, France).

To obtain supernatants and heat-inactivated bacteria, cultures were centrifuged at 4000× *g* for 5 min and suspended in RPMI-1640 medium or DMEM with the addition of 10% FBS (Fetal bovine serum) and 1% of L-glutamine (Euroclone, Pero, Italy) and incubated for 1 h at 37 °C. After centrifugation at 6000× *g* for 10 min, supernatants were collected. Bacterial inactivation was performed by heating at 95 °C for 15 min.

The bacterial lysates were obtained after incubation of bacterial cells with PBS 0.1% TRITON X-100 at 37 °C for 15 min. Before this step, Gram-positive bacteria were incubated with Lysozyme (10 mg/mL) for 1 h at 37 °C. Each sample was heated at 95 °C and immediately frozen in liquid nitrogen for at least three times. Protein concentration has been quantified by the BCA (Bicinchoninic Acid) method, and each sample was used as a stimulus at the concentration of 1 µg/mL.

### 4.3. Epithelial Cell Culture

SiHa cell line, isolated from squamous cell carcinoma and containing HPV-16 genome (1–2 copies per cell), was obtained from ATCC^®^ (ATCC^®^ HTB35™). SiHa cells were cultured in DMEM medium (Euroclone, Pero, Italy) supplemented with 10% FBS (Fetal bovine serum), 1% L-glutamine, 1% penicillin and streptomycin (Euroclone, Pero, Italy).

CaSki cell line, originally isolated from a cervical carcinoma and containing 600 copies of integrated HPV-16, was obtained from BEI-Resource. CaSki cells were cultured in RPMI-1640 medium (Euroclone, Pero, Italy) supplemented with 10% FBS (Fetal bovine serum), 1% L-glutamine, 1% penicillin and streptomycin (Euroclone, Pero, Italy).

### 4.4. PBMCs Isolation and Culture

Buffy coats from healthy donors (*n* = 9) were supplied by the Transfusional Center of Azienda Ospedaliera Careggi (Firenze, Italy). PBMCs were isolated by Ficoll-Paque density gradient (Cedarlane Labs, Burlington, Ontario, Canada) according to Paccosi et al. [53] and cultured in 6-well plates at the concentration of 10^6^ cells/mL in RPMI-1640 medium (Euroclone, Pero, Italy) supplemented with 10% FBS, 1% L-glutamine and 1% penicillin and streptomycin (Euroclone, Pero, Italy). After 1 h at 37 °C, cells were stimulated with heat-inactivated bacteria (50 MOI/cell) and cultured for additional 5 days at 37 °C and 5% di CO_2_. Culture supernatants from unstimulated or bacteria-stimulated cultures were collected, centrifuged at 4000 rpm for 10 min and stored at −80 °C.

### 4.5. Viability Test

SiHa and CaSki cells were cultured in 96-multiwells at the concentration of 2.5 × 10^3^ cells/mL with bacterial cell supernatants (10–20–30% *v/v*) or bacterial lysates (1 µg/mL). Intracellular ATP was measured as an index of metabolic activity using CellTiter-Glo^®^ LuminescentCell Viability Assay (Promega Corporation, Medison, WI, USA) after 24 h of culture.

### 4.6. Cytokines’ Evaluation

IFN-γ, IL-1β, IL-4, IL-6, IL-10, IL-17A, IP-10 and TNF-α cytokine’s concentration was measured using Milliplex^®^ Map kit Human Cytokine/Chemokine/Growth Factor Panel A Magnetic Bead Panel (Merck Kgaa, Darmstadt, Germany) and Luminex apparatus following manufacturer instruction (Luminex 200 MAGPIX).

IL-8 was measured by IL-8 Human ELISA (Thermo Fisher Scientific, Waltham, MA, USA) following the manufacturer’s instructions.

### 4.7. SCFAs (Short Chain Fait Acids) Profile of Bacterial Strains

Supernatants of bacterial strains were prepared from liquid cultures in broth containing TSB and 5% Horse lysed blood) (Oxoid, Basingstoke, UK) following OD determination and ultracentrifugation.

The SCFAs were assessed through an isotope dilution (ID) quantitative method [54] that uses gas chromatography coupled with a mass spectrometry (GC-MS) system.

Briefly, the SCFAs were recovered from the samples by liquid-liquid extraction and then analyzed, as free acid form, by GC-MS instrument equipped with a Supelco Nukol column 30 m length, 0.25 mm internal diameter and 0.25 µm of film thickness. The SCFAs separation was carried out by the temperatures program as follows: initial temperature of 40 °C was held for 1 min, then it was increased to 150 °C at 30 °C/min, finally grow up to 220 °C at 20 °C/min. A 1 µL aliquot of extracted sample was injected in spitless mode (spitless time 1 min) at 250 °C, while the transfer line temperature was 280 °C. The carrier flow rate was maintained at 1 mL/min.

The quantitative SCFAs’ evaluation was carried out by ratios between the area abundances of the analytes with the area abundances of respective labelled internal standards (isotopic dilution method). The ionic signals and the reference internal standard, used for the quantitation of each SCFA, were reported in Table 2. 3 mL of prefermented medium sample was added of 50μL of internal standards (ISTD) mixture, 1 mL of tert-butyl methyl ether and 50 µL of 1 M HCl solution in 15 mL centrifuge tube. Then, each tube was shaken in vortex apparatus for 2 min, centrifuged at 10,000 rpm for 5 min, and finally, the solvent layer was transferred in autosampler vial and analyzed by GC-MS method. Each sample has been prepared and processed three times by the previously described method.

### 4.8. Statistical Analysis

Numerical data were expressed as Mean ± standard deviation (SD) if they were in a normal distribution, or median and interquartile range (IQR) if they were not in Gaussian distribution. Mann–Whitney U test, Wilcoxon rank sum test, or Student t-test for two-group comparison was used, whereas ANOVA or Kruskal–Wallis with Bonferroni and Holm–Bonferroni p-value correction was used in case of multiple groups comparisons. Spearman rank correlation coefficient was used to examine the relationship between two continuous variables. Statistical significance was defined as a *p*-value < 0.05. Statistical analysis was performed using R software version 4.0.5. R Core Team (2021). R: A language and environment for statistical computing. R Foundation for Statistical Computing, Vienna, Austria. URL https://www.R-project.org/ (accessed on 31 March 2021).

## 5. Conclusions

Our results defined in more detail the contribution of each individual species of vaginal microbiota to the host’s defense against viral infection and revealed that not all vaginal lactobacilli have comparable properties of stimulating an adequate immune response. Based on the support of IFN-γ and lack of T_H_17 differentiation, *L. gasseri* appears to be the species that better assist the host’s defenses against HR-HPV infection.

## Figures and Tables

**Figure 1 ijms-22-06487-f001:**
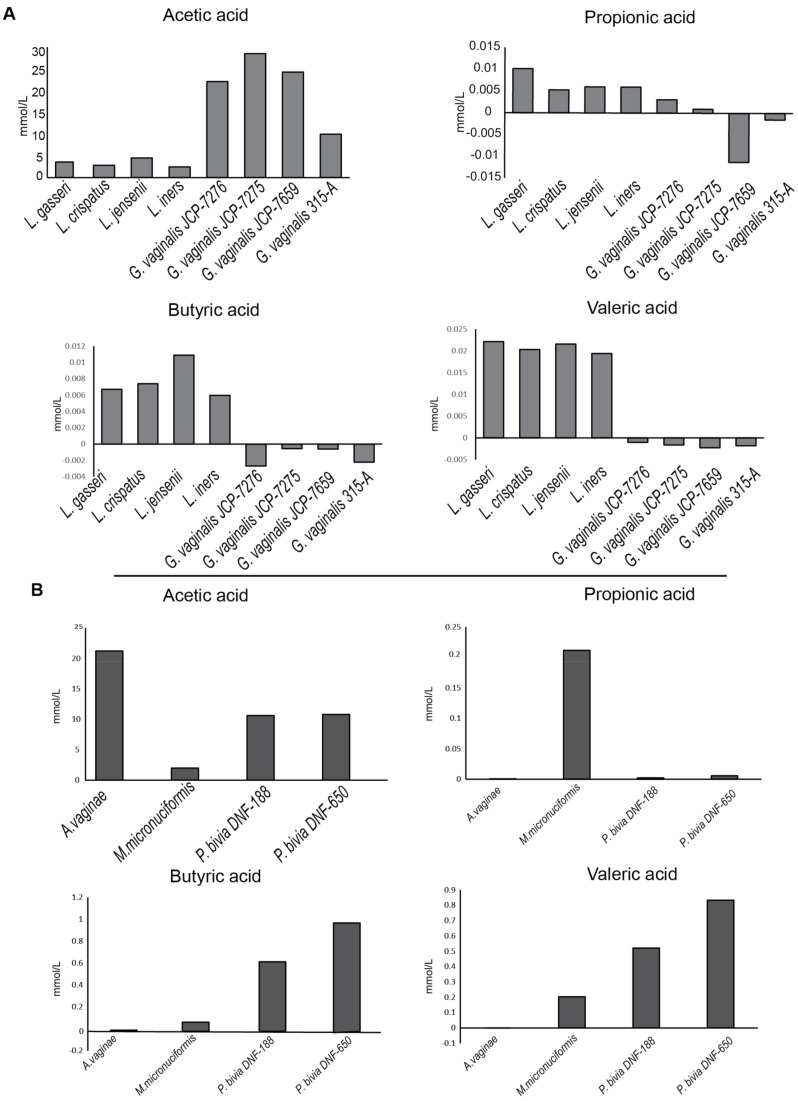
Short Chain Fatty Acids (SCFAs) profile. Supernatants of (**A**) *Lactobacillus* spp., *Gardnerella* spp. and (**B**) other vaginal dysbiosis bacteria were analyzed to the qualitative and quantitative determination of acetic, propionic, butyric and valeric acids using gas-chromatography coupled with mass spectrometry (GC-MS) system. The quantitative SCFAs’ evaluation was carried out by ratios between the area abundances of the analytes with the area abundances of respective labelled internal standards (isotopic dilution method).

**Figure 2 ijms-22-06487-f002:**
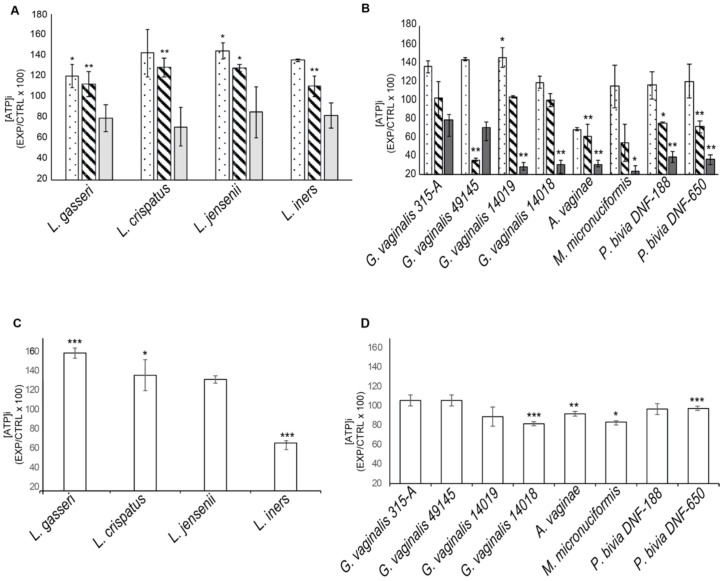
Effects of bacterial products on SiHa cell viability. (**A**,**B**) ATP production by cells were cultured with 10% (dotted-bar), 20% (ruled-bar), 30% (grey-bar) (*v/v*) of bacterial supernatants for 24 h. (**C**,**D**) ATP production by cells cultured with bacterial lysates (1 µg/mL) for 24 h. The bar-graph shows results from one representative experiment out of three performed. Data are expressed as ATP produced by stimulated cultures/unstimulated control ± standard deviation (EXP/CTRL × 100). Statistical analysis was performed by ANOVA and Student t-test. Significant differences among stimulated/ unstimulated cultures were reported * *p*-value < 0.05; ** *p*-value < 0.01; *** *p*-value < 0.001.

**Figure 3 ijms-22-06487-f003:**
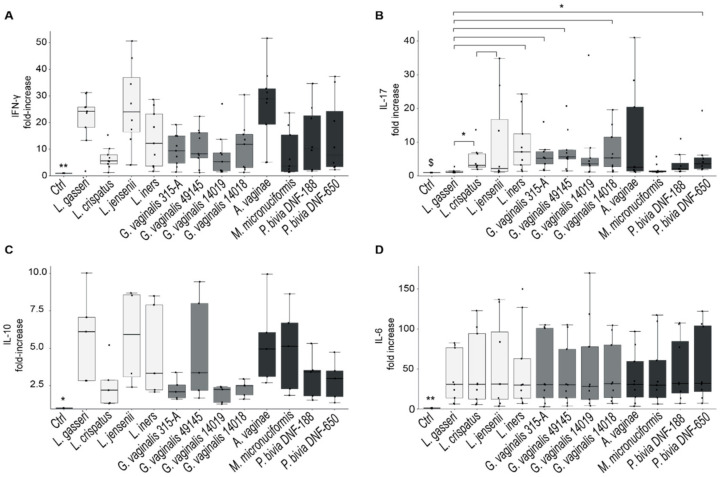
Cytokine’s production by PBMCs stimulated with dominant species of the vaginal microbiota. PBMCs were stimulated with 50 MOI/cell of heat-inactivated bacteria for 5 days. (**A**) IFN-γ, (**B**) IL-17, (**C**) IL-10 and (**D**) IL-6 concentration was measured in culture supernatants of bacterial-stimulated cultures. Results are expressed as a fold increase in cytokine concentration with respect to unstimulated cultures. The bar graph shows the median, and the whisker is calculated on the formula IQR × 1.5. Differences in cytokine concentrations among cultures were evaluated by Kruskal–Wallis test with Holm–Bonferroni p-value adjustment. ^$^ *p*-value referred to unstimulated cultures; * *p* < 0.05; ** *p* < 0.02.

**Figure 4 ijms-22-06487-f004:**
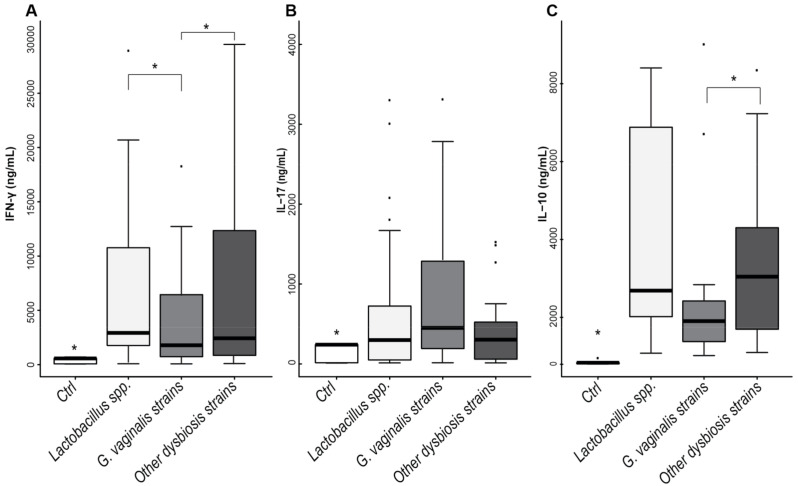
Cytokine production induced by lactobacilli or vaginal dysbiosis bacteria. Pooled bacterial lysates obtained by (1) *Lactobacillus* strains (*L. crispatus*, *L. jensenii*, *L. gasseri* and *L. iners*), (2) *G. vaginalis* strains (315-A, 49145, 14019 and 14018); (3) others vaginal dysbiosis bacteria (*A. vaginae*, *M. micronuciformis*, *P. bivia* DNF-188 and *P. bivia* DNF-650) were used as a stimulus for PBMCs (*n* = 9). (**A**) IFN-γ, (**B**) IL-17 and (**C**) IL-10 concentration was measured in culture supernatants of bacterial-stimulated cultures. The bar graph shows the median, and the whisker is calculated on the formula IQR × 1.5. Statistical analysis was performed by Kruskal–Wallis test, * *p* ≤ 0.01. Bonferroni p-value adjustment was used for IFN-γ and IL-10. Steel p-value adjustment was used for IL-17. Student t-test paired was used for IFN-γ and IL-10, *p* < 0.02.

**Figure 5 ijms-22-06487-f005:**
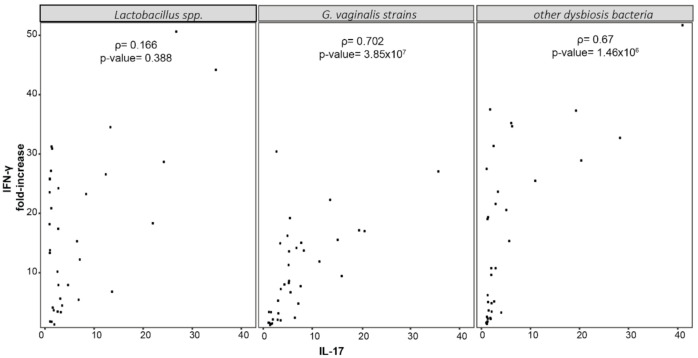
Spearman’s correlation analysis among cytokines produced in cultures with *Lactobacillus* spp., *G. vaginalis* strains, other vaginal dysbiosis bacteria. Correlation among cytokines produced under different stimulation was evaluated by Spearman rank correlation analysis.

**Table 1 ijms-22-06487-t001:** Reference bacterial strains.

CST	Family and Genus	Specie	Strain’s Name
I	*Lactobacillaceae*, *Lactobacillus*	*L. crispatus*	JV-V01
II	*Lactobacillaceae*, *Lactobacillus*	*L. gasseri*	SV-16A
III	*Lactobacillaceae*, *Lactobacillus*	*L. iners*	UPII-60-B
V	*Lactobacillaceae*, *Lactobacillus*	*L. jenseni*	JV-V16
IV	*Bifidobacteriaceae, Gardnerella*	*G. vaginalis*	315-A
IV	*Bifidobacteriaceae, Gardnerella*	*G. vaginalis*	49145/JCP-7276
IV	*Bifidobacteriaceae, Gardnerella*	*G. vaginalis*	14019/JCP-7659
IV	*Bifidobacteriaceae, Gardnerella*	*G. vaginalis*	14018/JCP-7275
IV	*Atopobiaceae, Atopobium*	*A. vaginalis*	DSM-15829
IV	*Prevotellaceae, Prevotella*	*P. bivia*	DNF-00188
IV	*Prevotellaceae, Prevotella*	*P. bivia*	DNF-00650
IV	*Veillonellaceae, Megasphaera*	*M. micronuciformis*	DNF-00954

**Table 2 ijms-22-06487-t002:** The ionic signal used for quali/quantitation and relative ISTD of each Short Chain Fat Acids (SCFAs) acquired by the ID-GC-MS method.

SCFAs	Quan. Ion	Qual. Ion	ISTD
Acetic acid	60	-	[^2^H3] Acetic
Propionic acid	74	73	[^2^H3] Propionic
Butyric acid	60	73	[^2^H3] Propionic
Valeric acid	60	73	[^2^H9] iso-Valeric

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
