# Peer review of "Vaginal Lactobacilli and Vaginal Dysbiosis-Associated Bacteria Differently Affect Cervical Epithelial and Immune Homeostasis and Anti-Viral Defenses"

_ijms, 2021, doi:10.3390/ijms22126487_

Round 1
Reviewer 1 Report
This paper describes data from research evaluating the effects of vaginal Lactobacilli and vaginal dysbiosis-associated bacteria on the production of inflammatory cytokines.
As I read this manuscript, I wrote down questions and suggestions, but at some point, the experimental design itself was questioned. If there is a problem with the design, all of the great experiments the authors have done are meaningless, so I'll go back to the beginning and replace it with simple questions.
- Why TSB with 5% horse lysed blood medium and 7-15% CO2 + 0.1% O2 is the environment for all microbial cultures? Each microorganism changes their metabolism dramatically according to the environment in which they are placed. For example, if minimal glucose based media with 5% horse lysed blood medium is used, the production pattern of Lactobacillus SCFAs will change significantly. How would tryptic soy broth represent the vaginal environment?
- Comparing cell stimulation effects with cell supernatants or cell lysates is an oversimplified experiment. Shouldn't the absolute amount of strains or their occupancy in the microbiota be considered in the actual vaginal environment?
Reviewer 2 Report
The manuscript analysing the variability of the microbiome on female reproductive tract is of growing interest.
The Authors suggest that a vaginal microbiota dominated by lactobacilli, (L. gasseri or L. jensenii), assist immune cells to clear HPV infections.
The manuscript is well written, however some minors issues should be corrected before its publication:
Please see the file in attachment that highlights the aspects/comments to be improved. (Please note: to hide the reviewer's information, the editor has turned comments in the PDF version into texts)
line 14: "HR-HPVs" highlighted;
line 21: "peripheral blood (PBMCs)" highlighted;
line 35: "(L.)" should be removed;
line 36: "L. crispatus, L. gasseri, L. iners and L. jensenii": Please check the current names according to Zheng, J.; Wittouck, S.; Salvetti, E.; Franz, C.; Harris, H.; Mattarelli, P.; O'Toole, P.; Pot, B.; Vandamme, P.; Walter, J.; Watanabe, K.; Wuyts, S.; Felis, G.; Gänzle, M.; Lebeer, S. A taxonomic note on the genus Lactobacillus: Description of 23 novel genera, emended description of the genus Lactobacillus Beijerinck 1901, and union of Lactobacillaceae and Leuconostocaceae. Int J Syst Evol Microbiol. 2020, 70, 2782-2858. doi: 10.1099/ijsem.0.004107.
line 66-68: Please insert the role of SCFA in vaginal microbiome and its impact on health.
line 84-85: "Gardnerella vaginalis, Atopobium vaginae, Megasphaera 84 micronuciformis, Prevotella bivia were": Please use Italic letter.
figure1: Are needed Error bars? in all figures.
figure 5: "lactobacilli" highlighted;
line 194: "Lactobacilli" highlighted;
line 197-199: Please insert López-Moreno & Aguilera 2021;
line 275-277: Please insert a separate section as Conclusions;
table 1: "Genre": Genus or Genera
table 1: "ATCC Code" highlighted;
Reviewer 3 Report
This is an interesting manuscript appraising the issue whether vaginal microbiota dominated by lactobacilli, especially by L. gasseri or L. jensenii, may be able to assist immune cells to clear HPV infection. Several issues have to be addressed by the author sbefore this manuscript can be accepted for publication in the International Journal of Molecular Sciences.
First and foremost, the rules of microbial nomenclature should be followed diligently throughout the manusctipt. Currently, some names are first abbreviated, then later given in the full format, etc. Also, when you abbreviate a genus, always put the abbreviation after the first mention of it - e.g., Lactobacillus (L.).
Abbreviations should not be used without the explanation in the Abstract section. R software used should be cited properly: R Core Team (2020). R: A language and environment for statistical computing. R Foundation for Statistical Computing, Vienna, Austria.
Other recent papers tackling the same issue should be included in the Discussion section. Please consult:
Borgogna JC, Shardell MD, Santori EK, Nelson TM, Rath JM, Glover ED, Ravel J, Gravitt PE, Yeoman CJ, Brotman RM. The vaginal metabolome and microbiota of cervical HPV-positive and HPV-negative women: a cross-sectional analysis. BJOG. 2020 Jan;127(2):182-192. doi: 10.1111/1471-0528.15981.
Kumari S, Bhor VM. Association of cervicovaginal dysbiosis mediated HPV infection with cervical intraepithelial neoplasia. Microb Pathog. 2021 Mar;152:104780. doi: 10.1016/j.micpath.2021.104780.
Author Response
- Currently, some names are first abbreviated, then later given in the full format, etc. Also, when you abbreviate a genus, always put the abbreviation after the first mention of it - e.g., Lactobacillus ().
- Author’s response: Now the abbreviations have been corrected as suggested. All changes are highlighted in the text.
- Abbreviations should not be used without the explanation in the Abstract section.
- Author’s response: Abbreviations were reviewed as suggested.
- R software used should be cited properly: R Core Team (2020). R: A language and environment for statistical computing. R Foundation for Statistical Computing, Vienna, Austria.
- Author’s response: Citation has been corrected.
- Other recent papers tackling the same issue should be included in the Discussion section. Please consult: Borgogna JC, Shardell MD, Santori EK, Nelson TM, Rath JM, Glover ED, Ravel J, Gravitt PE, Yeoman CJ, Brotman RM. The vaginal metabolome and microbiota of cervical HPV-positive and HPV-negative women: a cross-sectional analysis. BJOG. 2020 Jan;127(2):182-192. doi: 10.1111/1471-0528.15981. Kumari S, Bhor VM. Association of cervicovaginal dysbiosis mediated HPV infection with cervical intraepithelial neoplasia. Microb Pathog. 2021 Mar; 152:104780. doi: 10.1016/j.micpath.2021.104780.
- Author’s response: We accepted this criticism. The suggested papers have been included in the Discussion section and highlighted in yellow. Main text was changed as follows:
- Lines 215-217: Microbial and the host metabolites in the host microenvironment may affect the course of sexually transmitted infections. (Borgogna et al).
- Lines 241-243: A metabolomic analysis performed on vaginal samples from HPV+ and HPV- women ascertained that the metabolome of vaginal dysbiosis bacteria clustered differently from Lactobacillus-dominated microbiota. (Borgogna et al).
- Lines 249-251: Impairment of the vaginal epithelial barrier, chronic inflammation, alterations of the metabolic signaling and of immune response are all involved in carcinogenesis. (S Kumari and VK Bhor).
- Lines 255-256: Moreover, prolonged inflammatory response and high secretion of exosomes in vaginal environment may promote the progress of intraepithelial lesions (S Kumari and VK Bhor).
- Author’s response: We accepted this criticism. The suggested papers have been included in the Discussion section and highlighted in yellow. Main text was changed as follows:
Round 2
Reviewer 1 Report
Despite author's faithful revision efforts, it doesn't seem to be the answers to the questions I asked in the first place. The experiments in this paper are good references, but they seem to be difficult to publish as articles.
Author Response
- Despite author's faithful revision efforts, it doesn't seem to be the answers to the questions I asked in the first place.
Author’s response: We are sorry for not having completely addressed your suggestions. In this work, experiments were designed to assess the in vitro effects of representative vaginal bacterial taxa. In vivo experiments that will allow us to assess the effect of microbial communities on the immune response will be the subject of future studies.
Round 3
Reviewer 1 Report
Thank you for your revision efforts. But I still can't agree with the argument that just your in vitro experiments are worthy of being published as papers.